# Semantically Decomposing the Latent Spaces of Generative Adversarial Networks

**Chris Donahue**
Department of Music
University of California, San Diego
cdonahue@ucsd.edu

**Zachary C. Lipton**
Carnegie Mellon University
Amazon AI
zlipton@cmu.edu

**Akshay Balsubramani**
Department of Genetics
Stanford University
abalsubr@stanford.edu

**Julian McAuley**
Department of Computer Science
University of California, San Diego
jmcauley@eng.ucsd.edu

## Abstract

We propose a new algorithm for training generative adversarial networks that jointly learns latent codes for both identities (e.g. individual humans) and observations (e.g. specific photographs). By fixing the identity portion of the latent codes, we can generate diverse images of the same subject, and by fixing the observation portion, we can traverse the manifold of subjects while maintaining contingent aspects such as lighting and pose. Our algorithm features a pairwise training scheme in which each sample from the generator consists of two images with a common identity code. Corresponding samples from the real dataset consist of two distinct photographs of the same subject. In order to fool the discriminator, the generator must produce pairs that are photorealistic, distinct, and appear to depict the same individual. We augment both the DCGAN and BEGAN approaches with Siamese discriminators to facilitate pairwise training. Experiments with human judges and an off-the-shelf face verification system demonstrate our algorithm's ability to generate convincing, identity-matched photographs.

## 1 Introduction

In many domains, a suitable generative process might consist of several stages. To generate a photograph of a product, we might wish to first sample from the space of products, and then from the space of photographs *of that product*. Given such disentangled representations in a multistage generative process, an online retailer might diversify its catalog, depicting products in a wider variety of settings. A retailer could also flip the process, imagining new products in a fixed setting. Datasets for such domains often contain many labeled *identities* with fewer *observations* of each (e.g. a collection of face portraits with thousands of people and ten photos of each). While we may know the identity of the subject in each photograph, we may not know the *contingent* aspects of the observation (such as lighting, pose and background). This kind of data is ubiquitous; given a set of commonalities, we might want to incorporate this structure into our latent representations.

Generative adversarial networks (GANs) learn mappings from latent codes $\mathbf{z}$ in some low-dimensional space $\mathcal{Z}$ to points in the space of natural data $\mathcal{X}$ (Goodfellow et al., 2014). They achieve this power through an adversarial training scheme pitting a generative model $G : \mathcal{Z} \mapsto \mathcal{X}$ against a discriminative model $D : \mathcal{X} \mapsto [0, 1]$ in a minimax game. While GANs are popular, owing to their ability to generate high-fidelity images, they do not, in their original form, explicitly disentangle the latent factors according to known commonalities.

**In this paper**, we propose Semantically Decomposed GANs (SD-GANs), which encourage a specified portion of the latent space to correspond to a known source of variation.[1,2] The technique

---

[1] Web demo: https://chrisdonahue.github.io/sdgan
[2] Source code: https://github.com/chrisdonahue/sdgan

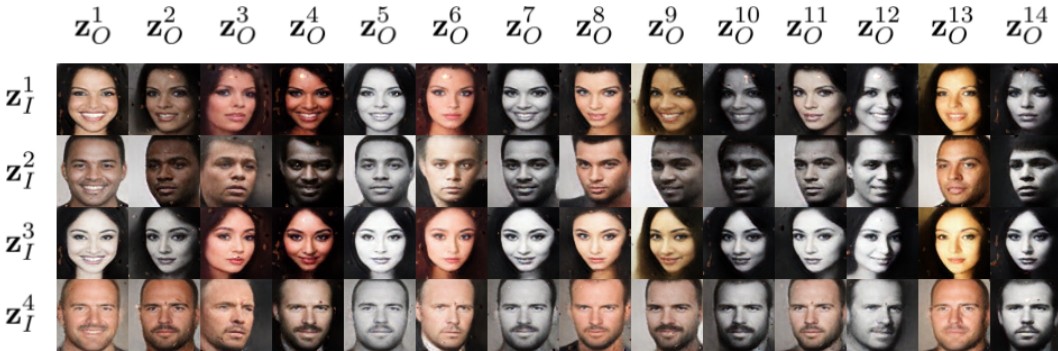

Figure 1: Generated samples from SD-BEGAN. Each of the four rows has the same identity code $\mathbf{z}_I$ and each of the fourteen columns has the same observation code $\mathbf{z}_O$.

decomposes the latent code $\mathcal{Z}$ into one portion $\mathcal{Z}_I$ corresponding to identity, and the remaining portion $\mathcal{Z}_O$ corresponding to the other contingent aspects of observations. SD-GANs learn through a pairwise training scheme in which each sample from the real dataset consists of two distinct images with a common identity. Each sample from the generator consists of a pair of images with common $\mathbf{z}_I \in \mathcal{Z}_I$ but differing $\mathbf{z}_O \in \mathcal{Z}_O$. In order to fool the discriminator, the generator must not only produce diverse and photorealistic images, but also images that depict the same identity when $\mathbf{z}_I$ is fixed. For SD-GANs, we modify the discriminator so that it can determine whether a pair of samples constitutes a match.

As a case study, we experiment with a dataset of face photographs, demonstrating that SD-GANs can generate contrasting images of the same subject (Figure 1; interactive web demo in footnote on previous page). The generator learns that certain properties are free to vary across observations but not identity. For example, SD-GANs learn that pose, facial expression, hirsuteness, grayscale vs. color, and lighting can all vary across different photographs of the same individual. On the other hand, the aspects that are more salient for facial verification remain consistent as we vary the observation code $\mathbf{z}_O$. We also train SD-GANs on a dataset of product images, containing multiple photographs of each product from various perspectives (Figure 4).

We demonstrate that SD-GANs trained on faces generate stylistically-contrasting, identity-matched image pairs that human annotators and a state-of-the-art face verification algorithm recognize as depicting the same subject. On measures of identity coherence and image diversity, SD-GANs perform comparably to a recent conditional GAN method (Odena et al., 2017); SD-GANs can also imagine new identities, while conditional GANs are limited to generating existing identities from the training data.

## 2    SEMANTICALLY DECOMPOSED GENERATIVE ADVERSARIAL NETWORKS

Before introducing our algorithm, we briefly review the prerequisite concepts.

### 2.1    GAN PRELIMINARIES

GANs leverage the discriminative power of neural networks to learn generative models. The generative model $G$ ingests latent codes $\mathbf{z}$, sampled from some known prior $P_{\mathcal{Z}}$, and produces $G(\mathbf{z})$, a sample of an implicit distribution $P_G$. The learning process consists of a minimax game between $G$, parameterized by $\theta_G$, and a discriminative model $D$, parameterized by $\theta_D$. In the original formulation, the discriminative model tries to maximize log likelihood, yielding

$$\min_G \max_D V(G, D) = \mathbb{E}_{\mathbf{x} \sim P_R}[\log D(\mathbf{x})] + \mathbb{E}_{\mathbf{z} \sim P_{\mathcal{Z}}}[\log(1 - D(G(\mathbf{z})))]. \qquad (1)$$

Training proceeds as follows: For $k$ iterations, sample one minibatch from the real distribution $P_R$ and one from the distribution of generated images $P_G$, updating discriminator weights $\theta_D$ to increase $V(G, D)$ by stochastic gradient ascent. Then sample a minibatch from $P_{\mathcal{Z}}$, updating $\theta_G$ to decrease $V(G, D)$ by stochastic gradient descent.

---

**Algorithm 1** Semantically Decomposed GAN Training

---
1: **for** n in 1:NumberOfIterations **do**
2:     **for** m in 1:MinibatchSize **do**
3:         Sample one identity vector $\mathbf{z}_I \sim \text{Uniform}([-1,1]^{d_I})$.
4:         Sample two observation vectors $\mathbf{z}_O^1, \mathbf{z}_O^2 \sim \text{Uniform}([-1,1]^{d_O})$.
5:         $\mathbf{z}^1 \leftarrow [\mathbf{z}_I; \mathbf{z}_O^1]$, $\mathbf{z}^2 \leftarrow [\mathbf{z}_I; \mathbf{z}_O^2]$.
6:         Generate pair of images $G(\mathbf{z}^1), G(\mathbf{z}^2)$, adding them to the minibatch with label 0 (fake).
7:     **for** m in 1:MinibatchSize **do**
8:         Sample one identity $i \in \mathcal{I}$ uniformly at random from the real data set.
9:         Sample two images of $i$ without replacement $\mathbf{x}_1, \mathbf{x}_2 \sim P_R(\mathbf{x}|I = i)$.
10:        Add the pair to the minibatch, assigning label 1 (real).
11:    Update discriminator weights by $\theta_D \leftarrow \theta_D + \nabla_{\theta_D} V(G, D)$ using its stochastic gradient.
12:    Sample another minibatch of identity-matched latent vectors $\mathbf{z}^1, \mathbf{z}^2$.
13:    Update generator weights by stochastic gradient descent $\theta_G \leftarrow \theta_G - \nabla_{\theta_G} V(G, D)$.

---

Zhao et al. (2017b) propose energy-based GANs (EBGANs), in which the discriminator can be viewed as an energy function. Specifically, they devise a discriminator consisting of an autoencoder: $D(\mathbf{x}) = D_d(D_e(\mathbf{x}))$. In the minimax game, the discriminator's weights are updated to minimize the reconstruction error $\mathcal{L}(\mathbf{x}) = ||\mathbf{x} - D(\mathbf{x})||$ for real data, while maximizing the error $\mathcal{L}(G(\mathbf{z}))$ for the generator. More recently, Berthelot et al. (2017) extend this work, introducing Boundary Equilibrium GANs (BEGANs), which optimize the Wasserstein distance (reminiscent of Wasserstein GANs (Arjovsky et al., 2017)) between autoencoder loss distributions, yielding the formulation:

$$V_{BEGAN}(G, D) = \mathcal{L}(\mathbf{x}) - \mathcal{L}(G(\mathbf{z})). \tag{2}$$

Additionally, they introduce a method for stabilizing training. Positing that training becomes unstable when the discriminator cannot distinguish between real and generated images, they introduce a new hyperparameter $\gamma$, updating the value function on each iteration to maintain a desired ratio between the two reconstruction errors: $\mathbb{E}[\mathcal{L}(G(\mathbf{z}))] = \gamma \mathbb{E}[\mathcal{L}(\mathbf{x})]$. The BEGAN model produces what appear to us, subjectively, to be the sharpest images of faces yet generated by a GAN. In this work, we adapt both the DCGAN (Radford et al., 2016) and BEGAN algorithms to the SD-GAN training scheme.

## 2.2 SD-GAN FORMULATION

Consider the data's identity as a random variable $I$ in a discrete index set $\mathcal{I}$. We seek to learn a latent representation that conveniently decomposes the variation in the real data into two parts: 1) due to $I$, and 2) due to the other factors of variation in the data, packaged as a random variable $O$. Ideally, the decomposition of the variation in the data into $I$ and $O$ should correspond exactly to a decomposition of the latent space $\mathcal{Z} = \mathcal{Z}_I \times \mathcal{Z}_O$. This would permit convenient interpolation and other operations on the inferred subspaces $\mathcal{Z}_I$ and $\mathcal{Z}_O$.

A conventional GAN samples $I, O$ from their joint distribution. Such a GAN's generative model samples directly from an unstructured prior over the latent space. It does not disentangle the variation in $O$ and $I$, for instance by modeling conditional distributions $P_G(O \mid I = i)$, but only models their average with respect to the prior on $I$.

Our SD-GAN method learns such a latent space decomposition, partitioning the coordinates of $\mathcal{Z}$ into two parts representing the subspaces, so that any $\mathbf{z} \in \mathcal{Z}$ can be written as the concatenation $[\mathbf{z}_I; \mathbf{z}_O]$ of its identity representation $\mathbf{z}_I \in \mathbb{R}^{d_I} = \mathcal{Z}_I$ and its contingent aspect representation $\mathbf{z}_O \in \mathbb{R}^{d_O} = \mathcal{Z}_O$. SD-GANs achieve this through a pairwise training scheme in which each sample from the real data consists of $\mathbf{x}_1, \mathbf{x}_2 \sim P_R(\mathbf{x} \mid I = i)$, a pair of images with a common identity $i \in \mathcal{I}$. Each sample from the generator consists of $G(\mathbf{z}_1), G(\mathbf{z}_2) \sim P_G(\mathbf{z} \mid \mathcal{Z}_I = \mathbf{z}_I)$, a pair of images generated from a common identity vector $\mathbf{z}_I \in \mathcal{Z}_I$ but i.i.d. observation vectors $\mathbf{z}_O^1, \mathbf{z}_O^2 \in \mathcal{Z}_O$. We assign identity-matched pairs from $P_R$ the label 1 and $\mathbf{z}_I$-matched pairs from $P_G$ the label 0. The discriminator can thus learn to reject pairs for either of two primary reasons: 1) not photorealistic or 2) not plausibly depicting the same subject. See Algorithm 1 for SD-GAN training pseudocode.

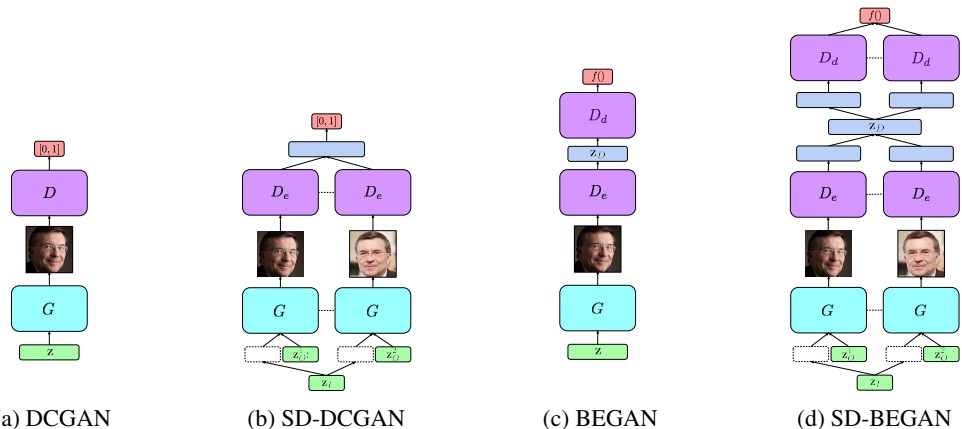

| (a) DCGAN | (b) SD-DCGAN | (c) BEGAN | (d) SD-BEGAN |
|---|---|---|---|

Figure 2: SD-GAN architectures and vanilla counterparts. Our SD-GAN models incorporate a decomposed latent space and Siamese discriminators. Dashed lines indicate shared weights. Discriminators also observe real samples in addition to those from the generator (not pictured for simplicity).

### 2.3 SD-GAN DISCRIMINATOR ARCHITECTURE

With SD-GANs, there is no need to alter the architecture of the generator. However, the discriminator must now act upon two images, producing a single output. Moreover, the effects of the two input images $\mathbf{x}_1, \mathbf{x}_2$ on the output score are not independent. Two images might be otherwise photorealistic but deserve rejection because they clearly depict different identities. To this end, we devise two novel discriminator architectures to adapt DCGAN and BEGAN respectively. In both cases, we first separately encode each image using the same convolutional neural network $D_e$ (Figure 2). We choose this Siamese setup (Bromley, 1994; Chopra et al., 2005) as our problem is symmetrical in the images, and thus it's sensible to share weights between the encoders.

To adapt DCGAN, we stack the feature maps $D_e(\mathbf{x}_1)$ and $D_e(\mathbf{x}_2)$ along the channel axis, applying one additional strided convolution. This allows the network to further aggregate information from the two images before flattening and fully connecting to a sigmoid output. For BEGAN, because the discriminator is an autoencoder, our architecture is more complicated. After encoding each image, we concatenate the representations $[D_e(\mathbf{x}_1); D_e(\mathbf{x}_2)] \in \mathbb{R}^{2(d_I+d_O)}$ and apply one fully connected bottleneck layer $\mathbb{R}^{2(d_I+d_O)} \Rightarrow \mathbb{R}^{d_I+2d_O}$ with linear activation. In alignment with BEGAN, the SD-BEGAN bottleneck has the same dimensionality as the tuple of latent codes $(\mathbf{z}_I, \mathbf{z}_O^1, \mathbf{z}_O^2)$ that generated the pair of images. Following the bottleneck, we apply a second FC layer $\mathbb{R}^{d_I+2d_O} \Rightarrow \mathbb{R}^{2(d_I+d_O)}$, taking the first $d_I + d_O$ components of its output to be the input to the first decoder and the second $d_I + d_O$ components to be the input to the second decoder. The shared intermediate layer gives SD-BEGAN a mechanism to push apart matched and unmatched pairs. We specify our exact architectures in full detail in Appendix E.

## 3 EXPERIMENTS

We experimentally validate SD-GANs using two datasets: 1) the *MS-Celeb-1M* dataset of celebrity face images (Guo et al., 2016) and 2) a dataset of shoe images collected from Amazon (McAuley et al., 2015). Both datasets contain a large number of identities (people and shoes, respectively) with multiple observations of each. The "in-the-wild" nature of the celebrity face images offers a richer test bed for our method as both identities and contingent factors are significant sources of variation. In contrast, Amazon's shoe images tend to vary only with camera perspective for a given product, making this data useful for sanity-checking our approach.

**Faces** From the aligned face images in the MS-Celeb-1M dataset, we select 12,500 celebrities at random and 8 associated images of each, resizing them to 64x64 pixels. We split the celebrities into subsets of 10,000 (training), 1,250 (validation) and 1,250 (test). The dataset has a small number of duplicate images and some label noise (images matched to the wrong celebrity). We detect and

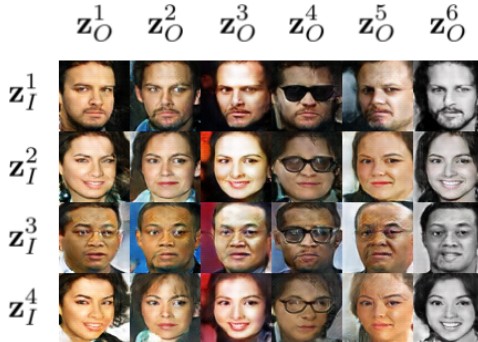

Figure 3: Generated samples from SD-DCGAN model trained on faces.

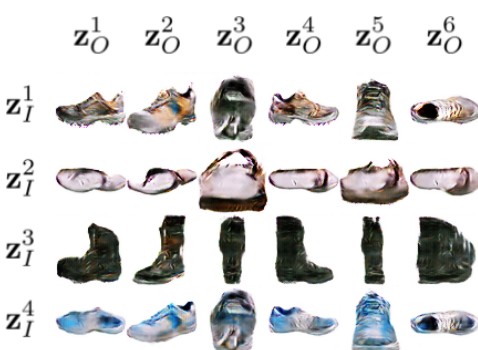

Figure 4: Generated samples from SD-DCGAN model trained on shoes.

remove duplicates by hashing the images, but we do not rid the data of label noise. We scale the pixel values to $[-1, 1]$, performing no additional preprocessing or data augmentation.

**Shoes**  Synthesizing novel product images is another promising domain for our method. In our shoes dataset, product photographs are captured against white backgrounds and primarily differ in orientation and distance. Accordingly, we expect that SD-GAN training will allocate the observation latent space to capture these aspects. We choose to study shoes as a prototypical example of a category of product images. The Amazon dataset contains around 3,000 unique products with the category "Shoe" and multiple product images. We use the same $80\%, 10\%, 10\%$ split and again hash the images to ensure that the splits are disjoint. There are 6.2 photos of each product on average.

## 3.1 Training details

We train SD-DCGANs on both of our datasets for 500,000 iterations using batches of 16 identity-matched pairs. To optimize SD-DCGAN, we use the Adam optimizer (Kingma & Ba, 2015) with hyperparameters $\alpha = 2e-4, \beta_1 = 0.5, \beta_2 = 0.999$ as recommended by Radford et al. (2016). We also consider a non-Siamese discriminator that simply stacks the channels of the pair of real or fake images before encoding (SD-DCGAN-SC).

As in (Radford et al., 2016), we sample latent vectors $\mathbf{z} \sim \text{Uniform}([-1, 1]^{100})$. For SD-GANs, we partition the latent codes according to $\mathbf{z}_I \in \mathbb{R}^{d_I}, \mathbf{z}_O \in \mathbb{R}^{100-d_I}$ using values of $d_I = [25, 50, 75]$. Our algorithm can be trivially applied with $k$-wise training (vs. pairwise). To explore the effects of using $k > 2$, we also experiment with an SD-DCGAN where we sample $k = 4$ instances each from $P_G(\mathbf{z} \mid \mathcal{Z}_I = \mathbf{z}_I)$ for some $\mathbf{z}_I \in \mathcal{Z}_I$ and from $P_R(\mathbf{x} \mid I = i)$ for some $i \in \mathcal{I}$. For all experiments, unless otherwise stated, we use $d_I = 50$ and $k = 2$.

We also train an SD-BEGAN on both of our datasets. The increased complexity of the SD-BEGAN model significantly increases training time, limiting our ability to perform more-exhaustive hyper-parameter validation (as we do for SD-DCGAN). We use the Adam optimizer with the default hyperparameters from (Kingma & Ba, 2015) for our SD-BEGAN experiments. While results from our SD-DCGAN $k = 4$ model are compelling, an experiment with a $k = 4$ variant of SD-BEGAN resulted in early mode collapse (Appendix F); hence, we excluded SD-BEGAN $k = 4$ from our evaluation.

We also compare to a DCGAN architecture trained using the auxiliary classifier GAN (AC-GAN) method (Odena et al., 2017). AC-GAN differs from SD-GAN in two key ways: 1) random identity codes $\mathbf{z}_I$ are replaced by a one-hot embedding over all the identities in the training set (matrix of size 10000x50); 2) the AC-GAN method encourages that generated photos depict the proper identity by tasking its discriminator with predicting the identity of the generated or real image. Unlike SD-GANs, the AC-DCGAN model cannot imagine new identities; when generating from AC-DCGAN (for our quantitative comparisons to SD-GANs), we must sample a random identity from those existing in the training data.

Table 1: Evaluation of 10k pairs from *MS-Celeb-1M* (real data) and generative models; half have matched identities, half do not. The *identity verification* metrics demonstrate that FaceNet (FN) and human annotators on Mechanical Turk (MT) verify generated data similarly to real data. The *sample diversity* metrics ensure that generated samples are statistically distinct in pixel space. Data generated by our best model (SD-BEGAN) performs comparably to real data. * 1k pairs, † 200 pairs.

| Dataset | Mem | Judge | Identity Verification | | | Sample Diversity | |
| | | | AUC | Acc. | FAR | ID-Div | All-Div |
|---|---|---|---|---|---|---|---|
| *MS-Celeb-1M* | - | FN | .913 | .867 | .045 | .621 | .699 |
| AC-DCGAN | 131 MB | FN | .927 | .851 | **.083** | .497 | .666 |
| SD-DCGAN | 57 MB | FN | .823 | .749 | .201 | .521 | .609 |
| SD-DCGAN-SC | **47** MB | FN | .831 | .757 | .180 | .560 | .637 |
| SD-DCGAN $k$=4 | 75 MB | FN | .852 | .776 | .227 | .523 | .614 |
| SD-DCGAN $d_I$=25 | 57 MB | FN | .835 | .764 | .222 | .526 | .615 |
| SD-DCGAN $d_I$=75 | 57 MB | FN | .816 | .743 | .268 | .517 | .601 |
| SD-BEGAN | 68 MB | FN | **.928** | **.857** | .110 | **.588** | **.673** |
| †*MS-Celeb-1M* | - | Us | - | .850 | .110 | .621 | .699 |
| *\*MS-Celeb-1M* | - | MT | - | .759 | .035 | .621 | .699 |
| *AC-DCGAN | 131 MB | MT | - | **.765** | **.090** | .497 | .666 |
| *SD-DCGAN $k$=4 | 75 MB | MT | - | .688 | .147 | .523 | .614 |
| *SD-BEGAN | **68** MB | MT | - | .723 | .096 | **.588** | **.673** |

## 3.2 EVALUATION

The evaluation of generative models is a fraught topic. Quantitative measures of sample quality can be poorly correlated with each other (Theis et al., 2016). Accordingly, we design an evaluation to match conceivable uses of our algorithm. Because we hope to produce diverse samples that humans deem to depict the same person, we evaluate the identity coherence of SD-GANs and baselines using both a pretrained face verification model and crowd-sourced human judgments obtained through Amazon's Mechanical Turk platform.

### 3.2.1 QUANTITATIVE

Recent advancements in face verification using deep convolutional neural networks (Schroff et al., 2015; Parkhi et al., 2015; Wen et al., 2016) have yielded accuracy rivaling humans. For our evaluation, we procure *FaceNet*, a publicly-available face verifier based on the Inception-ResNet architecture (Szegedy et al., 2017). The *FaceNet* model was pretrained on the CASIA-WebFace dataset (Yi et al., 2014) and achieves $98.6\%$ accuracy on the LFW benchmark (Huang et al., 2012).[3]

FaceNet ingests normalized, 160x160 color images and produces an embedding $f(\mathbf{x}) \in \mathbb{R}^{128}$. The training objective for FaceNet is to learn embeddings that minimize the $L_2$ distance between matched pairs of faces and maximize the distance for mismatched pairs. Accordingly, the embedding space yields a function for measuring the similarity between two faces $\mathbf{x}_1$ and $\mathbf{x}_2$: $D(\mathbf{x}_1, \mathbf{x}_2) = ||f(\mathbf{x}_1) - f(\mathbf{x}_2)||_2^2$. Given two images, $\mathbf{x}_1$ and $\mathbf{x}_2$, we label them as a match if $D(\mathbf{x}_1, \mathbf{x}_2) \leq \tau_v$ where $\tau_v$ is the accuracy-maximizing threshold on a class-balanced set of pairs from MS-Celeb-1M validation data. We use the same threshold for evaluating both real and synthetic data with FaceNet.

We compare the performance of FaceNet on pairs of images from the MS-Celeb-1M test set against generated samples from our trained SD-GAN models and AC-DCGAN baseline. To match FaceNet's training data, we preprocess all images by resizing from 64x64 to 160x160, normalizing each image individually. We prepare 10,000 pairs from MS-Celeb-1M, half identity-matched and half unmatched. From each generative model, we generate 5,000 pairs each with $\mathbf{z}_I^1 = \mathbf{z}_I^2$ and 5,000 pairs with $\mathbf{z}_I^1 \neq \mathbf{z}_I^2$. For each sample, we draw observation vectors $\mathbf{z}_O$ randomly.

We also want to ensure that identity-matched images produced by the generative models are diverse. To this end, we propose an intra-identity sample diversity (ID-Div) metric. The multi-scale structural similarity (MS-SSIM) (Wang et al., 2004) metric reports the similarity of two images on a scale from 0 (no resemblance) to 1 (identical images). We report 1 minus the mean MS-SSIM for all pairs

---

[3]"20170214-092102" pretrained model from `https://github.com/davidsandberg/facenet`

of identity-matched images as ID-Div. To measure the *overall* sample diversity (All-Div), we also compute 1 minus the mean similarity of 10k pairs with random identities.

In Table 1, we report the area under the receiver operating characteristic curve (AUC), accuracy, and false accept rate (FAR) of FaceNet (at threshold $\tau_v$) on the real and generated data. We also report our proposed diversity statistics. FaceNet verifies pairs from the real data with $87\%$ accuracy compared to $86\%$ on pairs from our SD-BEGAN model. Though this is comparable to the accuracy achieved on pairs from the AC-DCGAN baseline, our model produces samples that are more diverse in pixel space (as measured by ID-Div and All-Div). FaceNet has a higher but comparable FAR for pairs from SD-GANs than those from AC-DCGAN; this indicates that SD-GANs may produce images that are less *semantically* diverse on average than AC-DCGAN.

We also report the combined memory footprint of $G$ and $D$ for all methods in Table 1. For conditional GAN approaches, the number of parameters grows linearly with the number of identities in the training data. Especially in the case of the AC-GAN, where the discriminator computes a softmax over all identities, linear scaling may be prohibitive. While our 10k-identity subset of MS-Celeb-1M requires a 131MB AC-DCGAN model, an AC-DCGAN for all 1M identities would be over $8$GB, with more than $97\%$ of the parameters devoted to the weights in the discriminator's softmax layer. In contrast, the complexity of SD-GAN is constant in the number of identities.

### 3.2.2 QUALITATIVE

In addition to validating that identity-matched SD-GAN samples are verified by FaceNet, we also demonstrate that humans are similarly convinced through experiments using Mechanical Turk. For these experiments, we use balanced subsets of 1,000 pairs from MS-Celeb-1M and the most promising generative methods from our FaceNet evaluation. We ask human annotators to determine if each pair depicts the "same person" or "different people". Annotators are presented with batches of ten pairs at a time. Each pair is presented to three distinct annotators and predictions are determined by majority vote. Additionally, to provide a benchmark for assessing the quality of the Mechanical Turk ensembles, we (the authors) manually judged 200 pairs from MS-Celeb-1M. Results are in Table 1.

For all datasets, human annotators on Mechanical Turk answered "same person" less frequently than FaceNet when the latter uses the accuracy-maximizing threshold $\tau_v$. Even on real data, balanced so that $50\%$ of pairs are identity-matched, annotators report "same person" only $28\%$ of the time (compared to the $41\%$ of FaceNet). While annotators achieve higher accuracy on pairs from AC-DCGAN than pairs from SD-BEGAN, they also answer "same person" $16\%$ more often for AC-DCGAN pairs than real data. In contrast, annotators answer "same person" at the same rate for SD-BEGAN pairs as real data. This may be attributable to the lower sample diversity produced by AC-DCGAN. Samples from SD-DCGAN and SD-BEGAN are shown in Figures 3 and 1 respectively.

## 4 RELATED WORK

Style transfer and novel view synthesis are active research areas. Early attempts to disentangle style and content manifolds used factored tensor representations (Tenenbaum & Freeman, 1997; Vasilescu & Terzopoulos, 2002; Elgammal & Lee, 2004; Tang et al., 2013), applying their results to face image synthesis. More recent work focuses on learning hierarchical feature representations using deep convolutional neural networks to separate identity and pose manifolds for faces (Zhu et al., 2013; Reed et al., 2014; Zhu et al., 2014; Yang et al., 2015; Kulkarni et al., 2015; Oord et al., 2016; Yan et al., 2016) and products (Dosovitskiy et al., 2015). Gatys et al. (2016) use features of a convolutional network, pretrained for image recognition, as a means for discovering content and style vectors.

Since their introduction (Goodfellow et al., 2014), GANs have been used to generate increasingly high-quality images (Radford et al., 2016; Zhao et al., 2017b; Berthelot et al., 2017). Conditional GANs (cGANs), introduced by Mirza & Osindero (2014), extend GANs to generate class-conditional data. Odena et al. (2017) propose auxiliary classifier GANs, combining cGANs with a semi-supervised discriminator (Springenberg, 2015). Recently, cGANs have been used to ingest text (Reed et al., 2016) and full-resolution images (Isola et al., 2017; Liu et al., 2017; Zhu et al., 2017) as conditioning information, addressing a variety of image-to-image translation and style transfer tasks. Chen et al. (2016) devise an information-theoretic extension to GANs in which they maximize the mutual information between a subset of latent variables and the generated data. Their unsupervised method

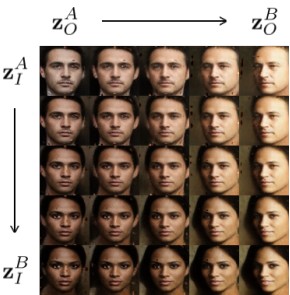 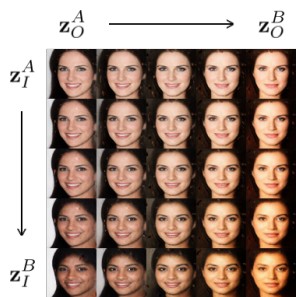 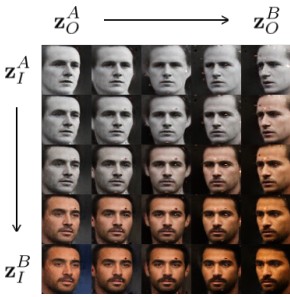

Figure 5: Linear interpolation of $\mathbf{z}_I$ (identity) and $\mathbf{z}_O$ (observation) for three pairs using SD-BEGAN generator. In each matrix, rows share $\mathbf{z}_I$ and columns share $\mathbf{z}_O$.

appears to disentangle some intuitive factors of variation, but these factors may not correspond to those explicitly disentangled by SD-GANs.

Several related papers use GANs for novel view synthesis of faces. Tran et al. (2017); Huang et al. (2017); Yin et al. (2017a;b); Zhao et al. (2017a) all address synthesis of different body/facial poses conditioned on an input image (representing identity) and a fixed number of pose labels. Antipov et al. (2017) propose conditional GANs for synthesizing artificially-aged faces conditioned on both a face image and an age vector. These approaches all require *explicit* conditioning on the relevant factor (such as rotation, lighting and age) in addition to an identity image. In contrast, SD-GANs can model these contingent factors implicitly (without supervision).

Mathieu et al. (2016) combine GANs with a traditional reconstruction loss to disentangle identity. While their approach trains with an encoder-decoder generator, they enforce a variational bound on the encoder embedding, enabling them to sample from the decoder without an input image. Experiments with their method only address small (28x28) grayscale face images, and their training procedure is complex to reproduce. In contrast, our work offers a simpler approach and can synthesize higher-resolution, color photographs.

One might think of our work as offering the generative view of the Siamese networks often favored for learning similarity metrics (Bromley, 1994; Chopra et al., 2005). Such approaches are used for discriminative tasks like face or signature verification that share the *many classes with few examples* structure that we study here. In our work, we adopt a Siamese architecture in order to enable the discriminator to differentiate between matched and unmatched pairs. Recent work by Liu & Tuzel (2016) propose a GAN architecture with weight sharing across multiple generators and discriminators, but with a different problem formulation and objective from ours.

## 5 DISCUSSION

Our evaluation demonstrates that SD-GANs can disentangle those factors of variation corresponding to identity from the rest. Moreover, with SD-GANs we can sample never-before-seen identities, a benefit not shared by conditional GANs. In Figure 3, we demonstrate that by varying the observation vector $\mathbf{z}_O$, SD-GANs can change the color of clothing, add or remove sunnies, or change facial pose. They can also perturb the lighting, color saturation, and contrast of an image, all while keeping the apparent identity fixed. We note, subjectively, that samples from SD-DCGAN tend to appear less photorealistic than those from SD-BEGAN. Given a generator trained with SD-GAN, we can independently interpolate along the identity and observation manifolds (Figure 5).

On the shoe dataset, we find that the SD-DCGAN model produces convincing results. As desired, manipulating $\mathbf{z}_I$ while keeping $\mathbf{z}_O$ fixed yields distinct shoes in consistent poses (Figure 4). The identity code $\mathbf{z}_I$ appears to capture the broad categories of shoes (sneakers, flip-flops, boots, etc.). Surprisingly, neither original BEGAN nor SD-BEGAN can produce diverse shoe images (Appendix G).

In this paper, we presented SD-GANs, a new algorithm capable of disentangling factors of variation according to known commonalities. We see several promising directions for future work. One logical extension is to disentangle latent factors corresponding to more than one known commonality. We also plan to apply our approach in other domains such as identity-conditioned speech synthesis.

ACKNOWLEDGEMENTS

The authors would like to thank Anima Anandkumar, John Berkowitz and Miller Puckette for their helpful feedback on this work. This work used the Extreme Science and Engineering Discovery Environment (XSEDE), which is supported by National Science Foundation grant number ACI-1053575 (Towns et al., 2014). GPUs used in this research were donated by the NVIDIA Corporation.

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

## A    ESTIMATING LATENT CODES

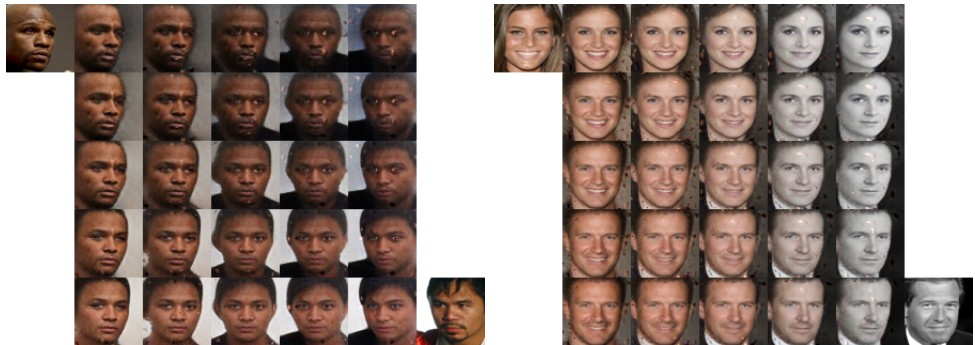

Figure 6: Linear interpolation of both identity (vertical) and observation (horizontal) on latent codes recovered for unseen images. All rows have the same identity vector ($\mathbf{z}_I$) and all columns have the same observation vector ($\mathbf{z}_O$).

We estimate latent vectors for unseen images and demonstrate that the disentangled representations of SD-GANs can be used to depict the estimated identity with different contingent factors. In order to find a latent vector $\hat{\mathbf{z}}$ such that $G(\hat{\mathbf{z}})$ (pretrained $G$) is similar to an unseen image $\mathbf{x}$, we can minimize the distance between $\mathbf{x}$ and $G(\hat{\mathbf{z}})$: $\min_{\hat{\mathbf{z}}} ||G(\hat{\mathbf{z}}) - \mathbf{x}||_2^2$ (Lipton & Tripathi, 2017).

In Figure 6, we depict estimation and linear interpolation across both subspaces for two pairs of images using SD-BEGAN. We also display the corresponding source images being estimated. For both pairs, $\hat{\mathbf{z}}_I$ (identity) is consistent in each row and $\hat{\mathbf{z}}_O$ (observation) is consistent in each column.

## B    PAIRWISE DISCRIMINATION OF EMBEDDINGS AND ENCODINGS

In Section 3.1, we describe an AC-GAN (Odena et al., 2017) baseline which uses an embedding matrix over real identities as latent identity codes ($G : i, \mathbf{z}_O \mapsto \hat{\mathbf{x}}$). In place of random identity vectors, we tried combining this identity representation with pairwise discrimination (in the style of SD-GAN). In this experiment, the discriminator receives either either two real images with the same identity $(\mathbf{x}_i^1, \mathbf{x}_i^2)$, or a real image with label $i$ and synthetic image with label $i$ $(\mathbf{x}_i^1, G(i, \mathbf{z}_O))$. All other hyperparameters are the same as in our SD-DCGAN experiment (Section 3.1). We show results in Figure 7.

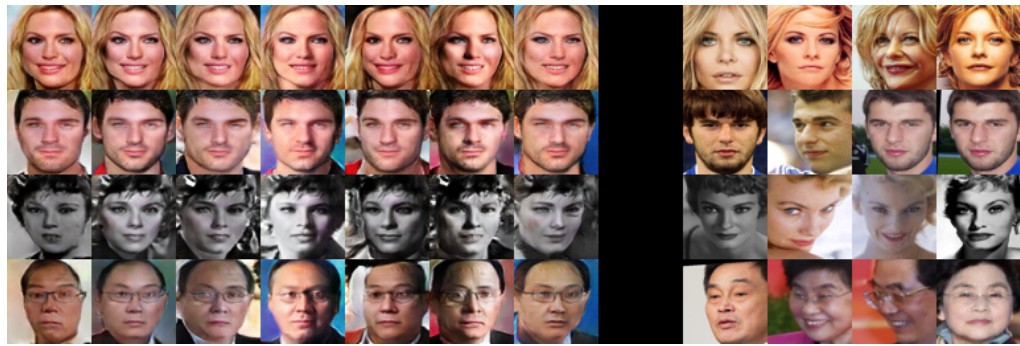

Figure 7: Generator with a one-hot identity embedding trained against a pairwise discriminator. Each row shares an identity vector and each column shares an observation vector. Random sample of $4$ real images of the corresponding identity on the right.

In Appendix C, we detail a modification of the DR-GAN (Tran et al., 2017) method which uses an encoding network $G_e$ to transform images to identity representations ($G_d : G_e(\mathbf{x}), \mathbf{z}_O \mapsto \hat{\mathbf{x}}$). We also tried combining this encoder-decoder approach with pairwise discrimination. The discriminator

receives either two real images with the same identity $(\mathbf{x}_i^1, \mathbf{x}_i^2)$, or $(\mathbf{x}_i^1, G_d(G_e(\mathbf{x}_i^1), \mathbf{z}_O)$. We show results in Figure 8.

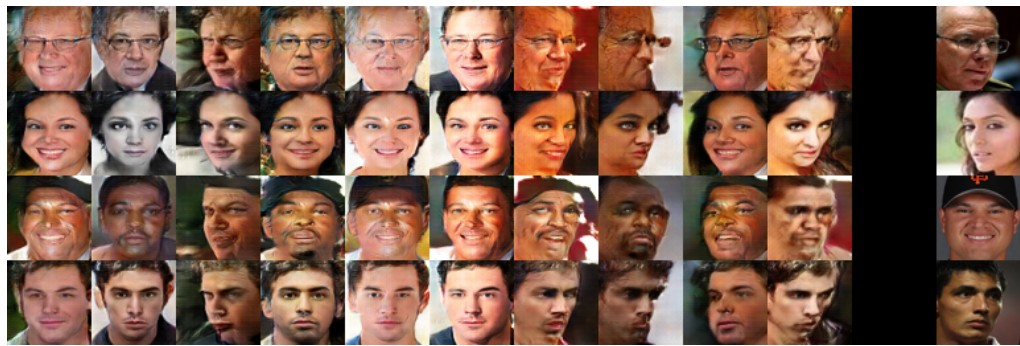

Figure 8: Generator with an encoder-decoder architecture trained against a pairwise discriminator. Each row shares an identity vector and each column shares an observation vector. Input image on the right.

While these experiments are exploratory and not part of our principle investigation, we find the results to be qualitatively promising. We are not the first to propose pairwise discrimination with pairs of (real, real) or (real, fake) images in GANs (Pathak et al., 2016; Isola et al., 2017).

## C EXPLORATORY EXPERIMENT WITH DR-GANS

Tran et al. (2017) propose *Disentangled Representation learning-GAN* (DR-GAN), an approach to face frontalization with similar setup to our SD-GAN algorithm. The (single-image) DR-GAN generator $G$ (composition of $G_e$ and $G_d$) accepts an input image $\mathbf{x}$, a pose code $\mathbf{c}$, and a noise vector $\mathbf{z}$. The DR-GAN discriminator receives either $\mathbf{x}$ or $\hat{\mathbf{x}} = G_d(G_e(\mathbf{x}), \mathbf{c}, \mathbf{z})$. In the style of (Springenberg, 2015), the discriminator is tasked with determining not only if the image is real or fake, but also classifying the pose $\mathbf{c}$, suggesting a disentangled representation to the generator. Through their experiments, they demonstrate that DR-GAN can explicitly disentangle pose and illumination ($\mathbf{c}$) from the rest of the latent space ($G_e(\mathbf{x}); \mathbf{z}$).

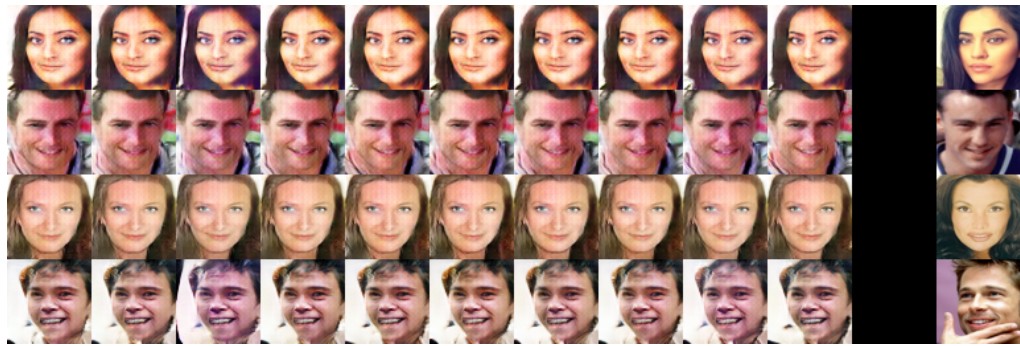

Figure 9: Generated samples from cGAN trained only to disentangle identity. Each row shares an identity vector and each column shares an observation vector; input image on the right.

In addition to our AC-DCGAN baseline (Odena et al., 2017), we tried modifying DR-GAN to only disentangle identity (rather than *both* identity and pose in the original paper). We used the DCGAN (Radford et al., 2016) discriminator architecture (Table 4) as $G_e$, linearly projecting the final convolutional layer to $G_e(\mathbf{x}) \in \mathbb{R}^{50}$ (in alignment with our SD-GAN experiments). We altered the discriminator to predict the identity of $\mathbf{x}$ or $\hat{\mathbf{x}}$, rather than pose information (which is unknown in our experimental setup). With these modifications, $G_e(\mathbf{x})$ is analogous to $\mathbf{z_I}$ in the SD-GAN generator, and $\mathbf{z}$ is analogous to $\mathbf{z_O}$. Furthermore, this setup is identical to the AC-DCGAN baseline

except that the embedding matrix is replaced by an encoding network $G_e$. Unfortunately, we found that the generator quickly learned to produce a single output image $\hat{x}$ for each input $x$ regardless of observation code $z$ (Figure 9). Accordingly, we excluded this experiment from our evaluation (Section 3.2).

## D    IMAGINING IDENTITIES WITH AC-GAN

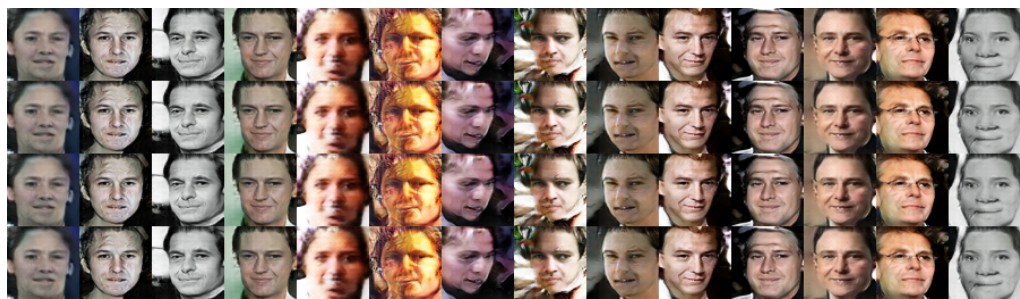

Figure 10: AC-DCGAN generation with random identity vectors that sum to one. Each row shares an identity vector and each column shares an observation vector.

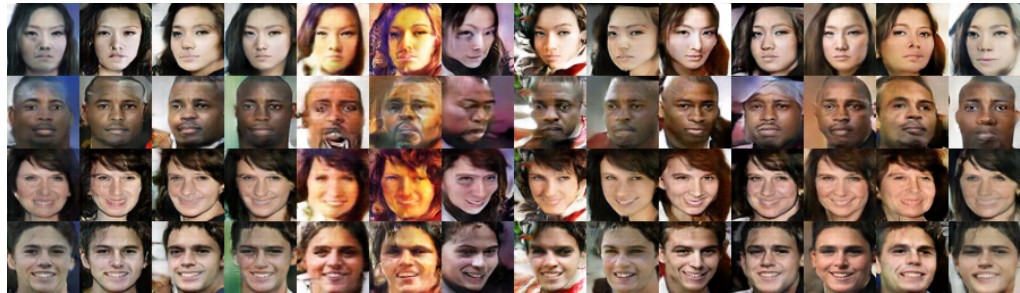

Figure 11: AC-DCGAN generation with one-hot identity vectors. Each row shares an identity vector and each column shares an observation vector.

As stated in Section 3.1, AC-GANs Odena et al. (2017) provide no obvious way to imagine new identities. For our evaluation (Section 3.2), the AC-GAN generator receives identity input $z_I \in [0, 1]^{10000}$: a one-hot over all identities. One possible approach to imagining *new* identities would be to query a trained AC-GAN generator with a random vector $z_I$ such that $\sum_{i=1}^{10000} z_I[i] = 1$. We found that this strategy produced little identity variety (Figure 10) compared to the normal one-hot strategy (Figure 11) and excluded it from our evaluation.

## E    ARCHITECTURE DESCRIPTIONS

We list here the full architectural details for our SD-DCGAN and SD-BEGAN models. In these descriptions, $k$ is the number of images that the generator produces and discriminator observes per identity (usually 2 for pairwise training), and $d_I$ is the number of dimensions in the latent space $\mathcal{Z}_I$ (identity). In our experiments, dimensionality of $\mathcal{Z}_O$ is always $100 - d_I$. As a concrete example, the bottleneck layer of the SD-BEGAN discriminator autoencoder ("fc2" in Table 6) with $k = 2, d_I = 50$ has output dimensionality 150.

We emphasize that generators are parameterized by $k$ in the tables only for clarity and symmetry with the discriminators. Implementations need not modify the generator; instead, $k$ can be collapsed into the batch size.

For the stacked-channels versions of these discriminators, we simply change the number of input image channels from 3 to $3k$ and set $k = 1$ wherever $k$ appears in the table.

Table 2: Input abstraction for both SD-DCGAN and SD-BEGAN generators during training (where $\mathbf{z}_O$ is always different for every pair or set of $k$)

| Operation | Input Shape | Kernel Size | Output Shape |
|---|---|---|---|
| $[\boldsymbol{z}_i; \boldsymbol{z}_o]$ | $[(d_I,);(k,100\text{-}d_I)]$ | | $[(d_I,);(k,100\text{-}d_I)]$ |
| dup $\boldsymbol{z}_i$ | $[(d_I,);(k,100\text{-}d_I)]$ | | $[(k,d_I);(k,100\text{-}d_I)]$ |
| concat | $[(k,d_I);(k,100\text{-}d_I)]$ | | $(k,100)$ |

Table 3: SD-DCGAN generator architecture

| Operation | Input Shape | Kernel Size | Output Shape |
|---|---|---|---|
| $\boldsymbol{z}$ | $(k,100)$ | | $(k,100)$ |
| fc1 | $(k,8192)$ | $(100,8192)$ | $(k,8192)$ |
| reshape | $(k,8192)$ | | $(k,4,4,512)$ |
| bnorm | $(k,4,4,512)$ | | $(k,4,4,512)$ |
| relu | $(k,4,4,512)$ | | $(k,4,4,512)$ |
| upconv1 | $(k,4,4,512)$ | $(5,5,512,256)$ | $(k,8,8,256)$ |
| bnorm | $(k,8,8,256)$ | | $(k,8,8,256)$ |
| relu | $(k,8,8,256)$ | | $(k,8,8,256)$ |
| upconv2 | $(k,8,8,256)$ | $(5,5,256,128)$ | $(k,16,16,128)$ |
| bnorm | $(k,16,16,128)$ | | $(k,16,16,128)$ |
| relu | $(k,16,16,128)$ | | $(k,16,16,128)$ |
| upconv3 | $(k,16,16,128)$ | $(5,5,128,64)$ | $(k,32,32,64)$ |
| bnorm | $(k,32,32,64)$ | | $(k,32,32,64)$ |
| relu | $(k,32,32,64)$ | | $(k,32,32,64)$ |
| upconv4 | $(k,32,32,64)$ | $(5,5,64,3)$ | $(k,64,64,3)$ |
| tanh | $(k,64,64,3)$ | | $(k,64,64,3)$ |

Table 4: SD-DCGAN discriminator architecture

| Operation | Input Shape | Kernel Size | Output Shape |
|---|---|---|---|
| $\boldsymbol{x}$ or $G(\boldsymbol{z})$ | $(k,64,64,3)$ | | $(k,64,64,3)$ |
| downconv1 | $(k,64,64,3)$ | $(5,5,3,64)$ | $(k,32,32,64)$ |
| lrelu(a=0.2) | $(k,32,32,64)$ | | $(k,32,32,64)$ |
| downconv2 | $(k,32,32,64)$ | $(5,5,64,128)$ | $(k,16,16,128)$ |
| bnorm | $(k,16,16,128)$ | | $(k,16,16,128)$ |
| lrelu(a=0.2) | $(k,16,16,128)$ | | $(k,16,16,128)$ |
| downconv3 | $(k,16,16,128)$ | $(5,5,128,256)$ | $(k,8,8,256)$ |
| bnorm | $(k,8,8,256)$ | | $(k,8,8,256)$ |
| lrelu(a=0.2) | $(k,8,8,256)$ | | $(k,8,8,256)$ |
| downconv4 | $(k,8,8,256)$ | $(5,5,256,512)$ | $(k,4,4,512)$ |
| stackchannels | $(k,4,4,512)$ | | $(4,4,512k)$ |
| downconv5 | $(4,4,512k)$ | $(3,3,512k,512)$ | $(2,2,512)$ |
| flatten | $(2,2,512)$ | | $(2048,)$ |
| fc1 | $(2048,)$ | $(2048,1)$ | $(1,)$ |
| sigmoid | $(1,)$ | | $(1,)$ |

Table 5: SD-BEGAN generator architecture

| Operation | Input Shape | Kernel Size | Output Shape |
|---|---|---|---|
| $z$ | $(k,100)$ | | $(k,100)$ |
| fc1 | $(k,100,)$ | $(100,8192)$ | $(k,100,8192)$ |
| reshape | $(k,100,8192)$ | | $(k,8,8,128)$ |
| conv2d | $(k,8,8,128)$ | $(3,3,128,128)$ | $(k,8,8,128)$ |
| elu | $(k,8,8,128)$ | | $(k,8,8,128)$ |
| conv2d | $(k,8,8,128)$ | $(3,3,128,128)$ | $(k,8,8,128)$ |
| elu | $(k,8,8,128)$ | | $(k,8,8,128)$ |
| upsample2 | $(k,8,8,128)$ | | $(k,16,16,128)$ |
| conv2d | $(k,16,16,128)$ | $(3,3,128,128)$ | $(k,16,16,128)$ |
| elu | $(k,16,16,128)$ | | $(k,16,16,128)$ |
| conv2d | $(k,16,16,128)$ | $(3,3,128,128)$ | $(k,16,16,128)$ |
| elu | $(k,16,16,128)$ | | $(k,16,16,128)$ |
| upsample2 | $(k,16,16,128)$ | | $(k,32,32,128)$ |
| conv2d | $(k,32,32,128)$ | $(3,3,128,128)$ | $(k,32,32,128)$ |
| elu | $(k,32,32,128)$ | | $(k,32,32,128)$ |
| conv2d | $(k,32,32,128)$ | $(3,3,128,128)$ | $(k,32,32,128)$ |
| elu | $(k,32,32,128)$ | | $(k,32,32,128)$ |
| upsample2 | $(k,32,32,128)$ | | $(k,64,64,128)$ |
| conv2d | $(k,64,64,128)$ | $(3,3,128,128)$ | $(k,64,64,128)$ |
| elu | $(k,64,64,128)$ | | $(k,64,64,128)$ |
| conv2d | $(k,64,64,128)$ | $(3,3,128,128)$ | $(k,64,64,128)$ |
| elu | $(k,64,64,128)$ | | $(k,64,64,128)$ |
| conv2d | $(k,64,64,128)$ | $(3,3,128,3)$ | $(k,64,64,3)$ |

Table 6: SD-BEGAN discriminator autoencoder architecture. The decoder portion is equivalent to, but **does not share weights with**, the SD-BEGAN generator architecture (Table 5).

| Operation | Input Shape | Kernel Size | Output Shape |
|---|---|---|---|
| $x$ or $G(z)$ | $(k,64,64,3)$ | | $(k,64,64,3)$ |
| conv2d | $(k,64,64,3)$ | $(3,3,3,128)$ | $(k,64,64,128)$ |
| elu | $(k,64,64,128)$ | | $(k,64,64,128)$ |
| conv2d | $(k,64,64,128)$ | $(3,3,128,128)$ | $(k,64,64,128)$ |
| elu | $(k,64,64,128)$ | | $(k,64,64,128)$ |
| conv2d | $(k,64,64,128)$ | $(3,3,128,128)$ | $(k,64,64,128)$ |
| elu | $(k,64,64,128)$ | | $(k,64,64,128)$ |
| downconv2d | $(k,64,64,128)$ | $(3,3,128,256)$ | $(k,32,32,256)$ |
| elu | $(k,32,32,256)$ | | $(k,32,32,256)$ |
| conv2d | $(k,32,32,256)$ | $(3,3,256,256)$ | $(k,32,32,256)$ |
| elu | $(k,32,32,256)$ | | $(k,32,32,256)$ |
| conv2d | $(k,32,32,256)$ | $(3,3,256,256)$ | $(k,32,32,256)$ |
| elu | $(k,32,32,256)$ | | $(k,32,32,256)$ |
| downconv2d | $(k,32,32,256)$ | $(3,3,256,384)$ | $(k,16,16,384)$ |
| elu | $(k,16,16,384)$ | | $(k,16,16,384)$ |
| conv2d | $(k,16,16,384)$ | $(3,3,384,384)$ | $(k,16,16,384)$ |
| elu | $(k,16,16,384)$ | | $(k,16,16,384)$ |
| conv2d | $(k,16,16,384)$ | $(3,3,384,384)$ | $(k,16,16,384)$ |
| elu | $(k,16,16,384)$ | | $(k,16,16,384)$ |
| downconv2d | $(k,16,16,384)$ | $(3,3,384,512)$ | $(k,8,8,512)$ |
| elu | $(k,8,8,512)$ | | $(k,8,8,512)$ |
| conv2d | $(k,8,8,512)$ | $(3,3,512,512)$ | $(k,8,8,512)$ |
| elu | $(k,8,8,512)$ | | $(k,8,8,512)$ |
| conv2d | $(k,8,8,512)$ | $(3,3,512,512)$ | $(k,8,8,512)$ |
| elu | $(k,8,8,512)$ | | $(k,8,8,512)$ |
| flatten | $(k,8,8,512)$ | | $(k,32768)$ |
| fc1 | $(k,32768)$ | $(32768,100)$ | $(k,100)$ |
| concat | $(k,100)$ | | $(100k,)$ |
| fc2 | $(100k,)$ | $(100k,d_I+(100-d_I)k,)$ | $(d_I+(100-d_I)k,)$ |
| fc3 | $(d_I+(100-d_I)k,)$ | $(d_I+(100-d_I)k,100k,)$ | $(100k,)$ |
| split | $(100k,)$ | | $(k,100)$ |
| $G$ (Table 5) | $(k,100)$ | | $(k,64,64,3)$ |

# F  FACE SAMPLES

We present samples from each model reported in Table 1 for qualitative comparison. In each matrix, $\mathbf{z}_I$ is the same across all images in a row and $\mathbf{z}_O$ is the same across all images in a column. We draw identity and observation vectors randomly for these samples.

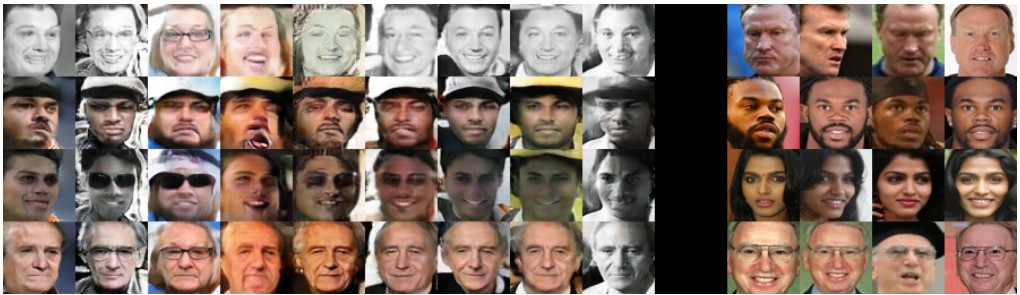

Figure 12: Generated samples from AC-DCGAN (four sample of real photos of ID on right)

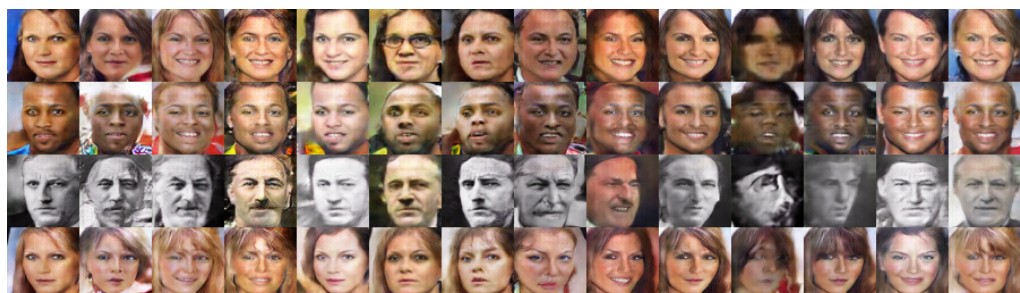

Figure 13: Generated samples from SD-DCGAN

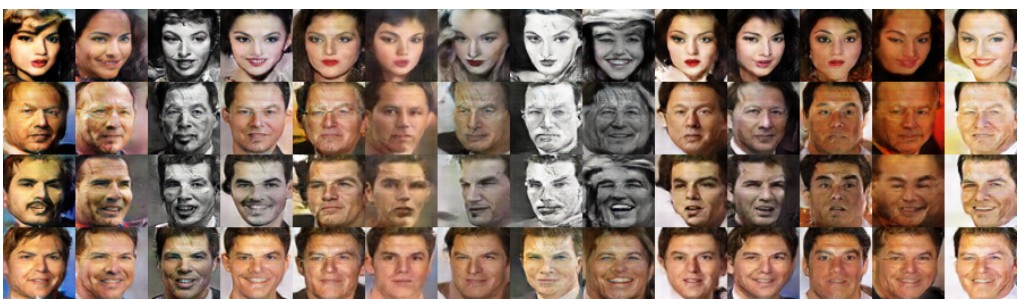

Figure 14: Generated samples from SD-DCGAN with stacked-channel discriminator

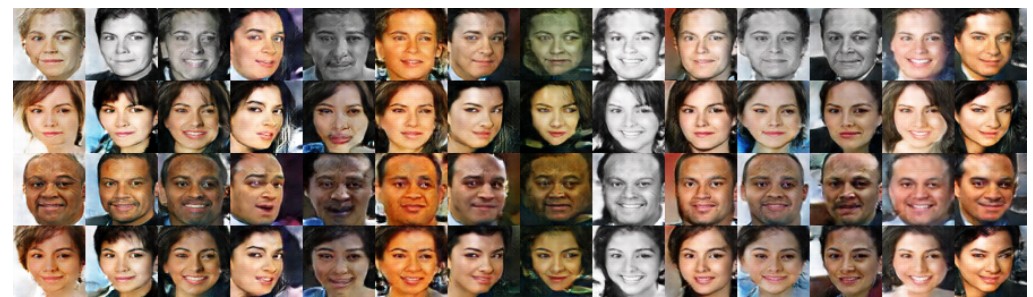

Figure 15: Generated samples from SD-DCGAN with $k = 4$

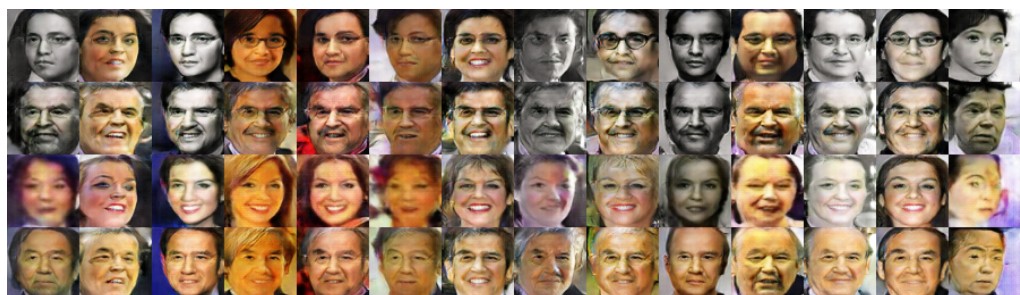

Figure 16: Generated samples from SD-DCGAN with $d_I = 25$

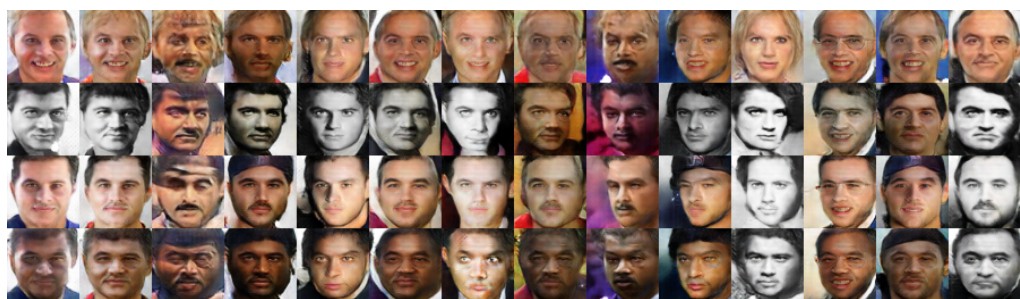

Figure 17: Generated samples from SD-DCGAN with $d_I = 75$

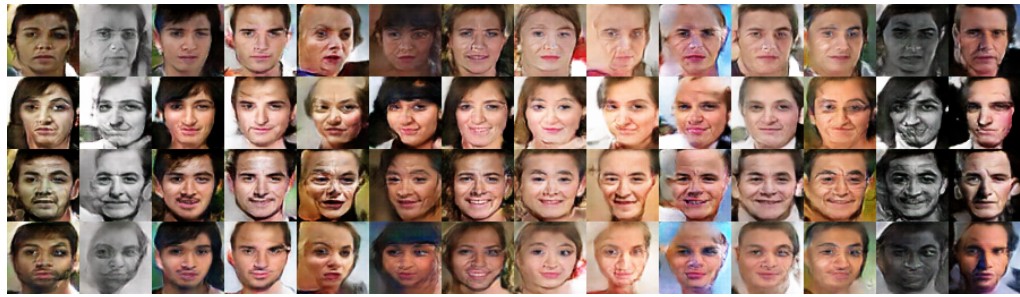

Figure 18: Generated samples from SD-DCGAN trained with the Wasserstein GAN loss (Arjovsky et al., 2017). This model was optimized using RMS-prop (Hinton et al.) with $\alpha = 5e-5$. In our evaluation (Section 3.2), FaceNet had an AUC of .770 and an accuracy of 68.5% (at $\tau_v$) on data generated by this model. We excluded it from Table 1 for brevity.

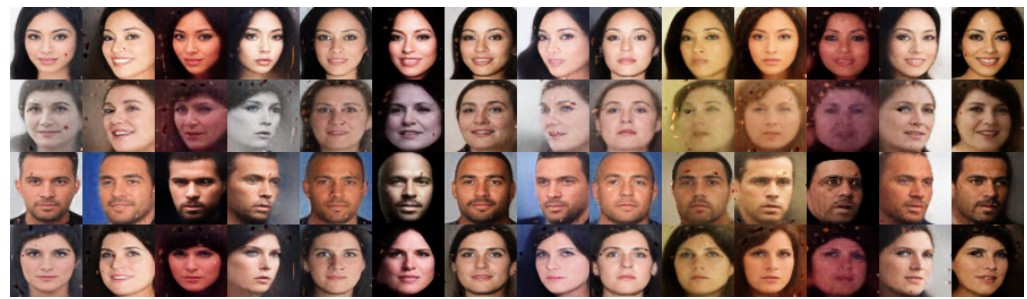

Figure 19: Generated samples from SD-BEGAN

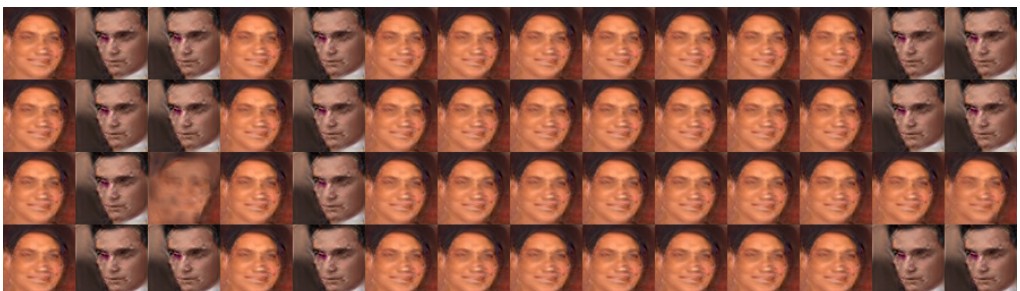

Figure 20: Generated samples from SD-BEGAN with $k = 4$, demonstrating mode collapse

## G  SHOE SAMPLES

We present samples from an SD-DCGAN and SD-BEGAN trained on our shoes dataset.

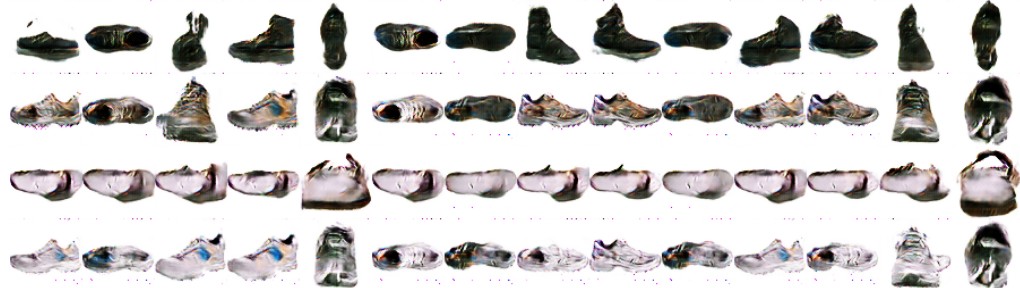

Figure 21: Generated samples from SD-DCGAN

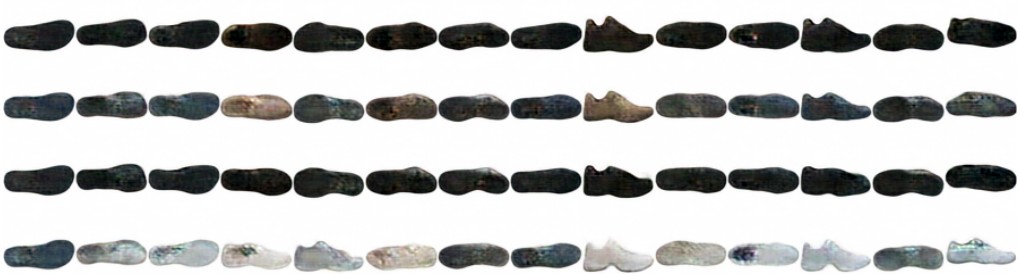

Figure 22: Generated samples from SD-BEGAN

