# OpenReview forum: "Semantically Decomposing the Latent Spaces of Generative Adversarial Networks"
_ICLR.cc/2018/Conference — Accept (Poster)_

### Official Review · AnonReviewer2 · 2017-11-25
**Progress in disentangling identity from other factors using GANs.**

**Rating:** 6
**Confidence:** 4

**Review:**

Quality
The paper is well written and the model is simple and clearly explained. The idea for disentangling identity from other factors of variation using identity-matched image pairs is quite simple, but the experimental results on faces and shoes are impressive.

Clarity
The model and its training objective are simple and clearly explained.

Originality
There are now many, many papers on generative models with disentangled feature representations, including with GANs. However, to my knowledge this is the first paper showing very compelling results using this particular setup of identity-aligned images.

Significance
Disentangled generative models are an important line of work in my opinion. This paper presents a very simple but apparently effective way of disentangling identity from other factors, and implements in two of the more recent GAN architectures.

Suggestion for an experiment - can you do few shot image generation? A simple way to do it would be to train an encoder from image → identity encoding. Then, given one or a few images of a new person’s face or a new shoe, you could estimate the identity latent variable, and then generate many additional samples.

Pros
- Very simple and effective disentangling technique for GANs.
- Great execution, compelling samples on both faces and shoes.

Cons
- Only two factors of variations are disentangled in this model. Could it be generalized to specify more than just two, e.g. lighting, pose, viewpoint, etc?
- Not much technically new or surprising compared to past work on disentangling generative models.

---

> ### Author Response · Authors · 2017-12-12
> **Response to AnonReviewer2**
>
>
> Thank you for your insights. We are glad that you found our paper to be well-written and our method to be both simple and effective. Your intuition is correct that SD-GAN could be extended to disentangle multiple factors of variation. We are actively investigating this and plan to publish these results in future work. You are also correct that SD-GAN could be modified for few-shot image generation; please see Appendix B (and the relevant Figure 8) for our preliminary investigation and results in this setting.

---

### Official Review · AnonReviewer3 · 2017-11-26
**Review summary of "Semantically Decomposing the Latent Spaces of Generative Adversarial Networks"**

**Rating:** 6
**Confidence:** 4

**Review:**

Summary:

This paper investigated the problem of controlled image generation. Assuming images can be disentangled by identity-related factors and style factors, this paper proposed an algorithm that produces a pair of images with the same identity. Compared to standard GAN framework, this algorithm first generated two latent variables for the pair images. The two latent variables are partially shared reflecting the shared identity information. The generator then transformed the latent variables into high-resolution images with a deconvolution decoder networks. The discriminator was used to distinguish paired images from database or paired images sampled by the algorithm. Experiments were conducted using DCGAN and BEGAN on portrait images and shoe product images. Qualitative results demonstrated that the learned style representations capture viewpoint, illumination and background color while the identity was well preserved by the identity-related representations.


== Novelty & Significance ==
Paired image generation is an interesting topic but this has been explored to some extent. Compared to existing coupled generation pipeline such as CoGAN, I can see the proposed formulation is more application-driven.

== Technical Quality ==
In Figure 3, the portrait images in the second row and fourth row look quite similar. I wonder if the trained model works with only limited variability (in terms of identity).
In Figure 4, the viewpoint is quite limited (only 4 viewpoints are provided).

I am not very convinced whether SD-GAN is a generic algorithm for controlled image generation. Based on the current results, I suspect it only works in fairly constrained settings.
It would be good to know if it actually works in more challenging datasets such as SUN bedroom, CUB and Oxford Flowers.

“the AC-DCGAN model cannot imagine new identities”
I feel the author of this paper made an unfair argument when comparing AC-DCGAN with the proposed method. First, during training, the proposed SD-GAN needs to access the identity information and there is only limited identity in the dataset. Based on the presentation, it is not very clear how does the model generate novel identities (in contrast to simply interpolating existing identities). For example, is it possible to generate novel viewpoints in Figure 4?

Missing references on conditional image generation and coupled image generation:
-- Generative Adversarial Text-to-Image Synthesis. Reed et al., In ICML 2016.
-- Attribute2Image: Conditional Image Generation from Visual Attributes. Yan et al., In ECCV 2016.
-- Domain Separation Networks. Bousmalis et al., In NIPS 2016.
-- Unsupervised Image-to-Image Translation Networks. Liu et al., In NIPS 2017.

Overall, I rate this paper slightly above borderline. It showed some good visualization results on controlled image generation. But the comparison to AC-GAN is not very fair, since the identity pairs are fully supervised for the proposed method. As far as I can see, there are no clear-cut improvements quantitatively. Also, there is no comparison with CoGAN, which I believe is the most relevant work for coupled image generation.

---

> ### Author Response · Authors · 2017-12-12
> **Response to AnonReviewer3**
>
>
> Thank you for your response and for pointing out missing references. We have added them to our related work section in the updated manuscript. Please see our general response above with regard to identity diversity.
>
> We did not quantitatively compare to CoGAN as their problem objectives are different from our own. CoGANs learn to translate images between domains with binary attributes such as blond/brown hair or glasses/no-glasses without parallel data. How one might extend CoGANs to learn a manifold over thousands of identities is non-obvious.
>
> As you suggested, we would also like to run studies on other types of data (the Oxford Flowers dataset that you recommended is particularly enticing), but leave this as an avenue for our future explorations.

---

### Official Review · AnonReviewer1 · 2017-11-27
**Review from AnonReviewer1**

**Rating:** 7
**Confidence:** 4

**Review:**

[Overview]

In this paper, the authors proposed a model called SD-GAN, to decompose semantical component of the input in GAN. Specifically, the authors proposed a novel architecture to decompose the identity latent code and non-identity latent code. In this new architecture, the generator is unchanged while the discriminator takes pair data as the input, and output the decision of whether two images are from the same identity or not. By training the whole model with a conventional GAN-training regime, SD-GAN learns to take a part of the input Z as the identity information, and the other part of input Z as the non-identity (or attribute) information. In the experiments, the authors demonstrate that the proposed SD-GAN could generate images preserving the same identity with diverse attributes, such as pose, age, expression, etc. Compared with AC-GAN, the proposed SD-GAN achieved better performance in both automatically evaluation metric (FaceNet) and Human Study. In the appendix, the authors further presented ablated qualitative results in various settings.

[Strengths]

1. This paper proposed a simple but effective generative adversarial network, called SD-GAN, to decompose the input latent code of GAN into separate semantical parts. Specifically, it is mainly instantiated on face images, to decompose the identity part and non-identity part in the latent code. Unlike the previous works such as AC-GAN, SD-GAN exploited a Siamese network to replace the conventional discriminator used in GAN. By this way, SD-GAN could generate images of novel identities, rather than being constrained to those identities used during training. I think this is a very good property. Due to this, SD-GAN consumes much less memory than AC-GAN, when training on a large number of identities.

2. In the experiment section, the authors quantitatively evaluate the generated images based on two methods, one is using a pre-trained FaceNet model to measure the verification accuracy and one is human study. When evaluated based on FaceNet, the proposed SD-GAN achieved higher accuracy and obtained more diverse face images, compared with AC-GAN. In human study, SD-GAN achieved comparable verification accuracy, while higher diversity than AC-GAN. The authors further presented ablated experiments in the Appendix.

[Comments]

This paper presents a novel model to decompose the latent code in a semantic manner. However, I have several questions about the model:

1. Why would SD-GAN not generate images merely have a smaller number of identities or just a few identities? In Algorithm 1, the authors trained the model by sampling one identity vector, which is then concatenated to two observation vectors. In this case, the generator always takes the same identity vectors, and the discriminator is used to distinguish these fake same-identity pair and the real same-identity pair from training data. As such, even if the generator generates the same identity, say mean identity, given different identity vectors, the generated images can still obtain a lower discrimination loss. Without any explicite constraint to enforce the generator to generate different identity with different identity vectors, I am wondering what makes SD-GAN be able to generate diverse identities?

2. Still about the identity diversity. Though the authors showed the identity-matched diversity in the experiments, the diversity across identity on the generated images is not evaluated. The authors should also evaluate this kind of identity. Generally, AC-GAN could generate as many identities as the number of identities in training data. I am curious about whether SD-GAN could generate comparable diverse identity to AC-GAN. One simple way is to evaluate the whole generated image set using Inception Score based on a Pre-trained face identification network; Another way is to directly use the generated images to train a verification model or identification model and evaluate it on real images. Though compared with AC-GAN, SD-GAN achieved better identity verification performance and sample diversity, I suspect the identity diversity is discounted, though SD-GAN has the property of generating novel identities. Furthermore,  the authors should also compare the general quality of generated samples with DC-GAN and BEGAN as well (at least qualitatively), apart from the comparison to AC-GAN on the identity-matched generation.

3. When making the comparison with related work, the authors mentioned that Info-GAN was not able to determine which factors are assigned to each dimension. I think this is not precise. The lack of this property is because there are no data annotations. Given the data annotations, Info-GAN can be easily augmented with such property by sending the real images into the discriminator for classification. Also, there is a typo in the caption of Fig. 10. It looks like each column shares the same identity vector instead of each row.

[Summary]

This paper proposed a new model called SD-GAN to decompose the input latent code of GAN into two separete semantical parts, one for identity and one for observations. Unlike AC-GAN, SD-GAN exploited a Siamese architecture in discriminator. By this way, SD-GAN could not only generate more identity-matched face image pairs but also more diverse samples with the same identity,  compared with AC-GAN. I think this is a good idea for decomposing the semantical parts in the latent code, in the sense that it can imagine new face identities and consumes less memory during training. Overall, I think this is a good paper. However, as I mentioned above, I am still not clear why SD-GAN could generate diverse identities without any constraints to make the model do that. Also, the authors should further evaluate the diversity of identity and compare it with AC-GAN.

---

> ### Author Response · Authors · 2017-12-12
> **Response to AnonReviewer1**
>
>
> Thank you for your detailed comments. In addition to adding information about identity diversity to the manuscript (see general response above), we offer the following responses to your other inquiries:
>
> 1) Why would SD-GAN not generate images merely have a smaller number of identities or just a few identities?
>
> If SD-GANs always produced images depicting the same identity regardless of the identity vector, the discriminator would be able to easily label these samples as fake on the basis that they always depict the same subject. This is the same reason that regular GANs are able to produce images that differ in subject identity.
>
> 3) When making the comparison with related work, the authors mentioned that Info-GAN was not able to determine which factors are assigned to each dimension. I think this is not precise.
>
> Thank you for pointing out that our statement about InfoGAN was imprecise. Our original purpose in making this claim was to indicate that there is no *obvious* way to hold identity fixed while varying contingent factors in vanilla InfoGAN, and hence no way to directly compare it to our SD-GAN algorithm. However, as you pointed out, InfoGAN is fully unsupervised while SD-GAN is not. We have updated our related work section to clarify the distinction between InfoGAN and SD-GAN.

---

### Author Response · Authors · 2017-12-12
**General response to reviews**

We would like to thank all of the reviewers for their thoughtful comments and suggestions. We are glad that reviewers found our method simple, our results compelling, and our paper to be well-written. While the overall response was positive, reviewers expressed minor concerns related to identity diversity in our generated results. We have uploaded a new version of the paper that will hopefully clarify.

[Identity diversity]

In the original manuscript, we report one measure of identity diversity. In Table 1, our All-Div metric reports the mean diversity (one minus MS-SSIM) for 10k pairs with random identity. While All-Div captures diversity at the pixel level, it is perhaps not the best measure of *semantic* (perceived) diversity. In our updated manuscript, we report another metric in Table 1: the false accept rate (FAR) of FaceNet and human annotators. A higher FAR indicates some evidence of lower identity diversity. By this metric, SD-GANs produce images with lower but comparable diversity to those from AC-DCGAN, and both have lower diversity than the real data. We have added these details to Table 1.

---

### Decision · Program_Chairs · 2018-01-29
**ICLR 2018 Conference Acceptance Decision**

**Decision:**

Accept (Poster)

**Comment:**

The paper proposes a GAN based approach for disentangling identity (or class information) from style. The supervision needed is the identity label for each image. Overall, the reviewers agree that the paper makes a novel contribution along the line of work on disentangling 'style' from 'content'.